# Somatic Mutation of the Non-Muscle-Invasive Bladder Cancer Associated with Early Recurrence

**DOI:** 10.3390/diagnostics13203201

**Published:** 2023-10-13

**Authors:** Seong Hyeon Yu, Sung Sun Kim, Hyungki Lee, Shinseung Kim, Taek Won Kang

**Affiliations:** 1Department of Urology, Chonnam National University Medical School, Chonnam National University Hospital, Gwangju 61469, Republic of Korea; domer12@hanmail.net; 2Department of Pathology, Chonnam National University Medical School, Chonnam National University Hospital, Gwangju 61469, Republic of Korea; kimsspathology@jnu.ac.kr; 3MediCloud Corporation, Hwasun 58128, Republic of Korea; hklee@medicloudgroup.com (H.L.); sskim@medicloudgroup.com (S.K.)

**Keywords:** next-generation sequencing, genetic testing, urinary bladder neoplasms, biomarkers

## Abstract

Next-generation sequencing (NGS) is widely used in muscle-invasive bladder cancer but has limited use in non-muscle-invasive bladder cancer (NMIBC) due to significant heterogeneity and high cancer-specific survival. Therefore, we evaluated the genomic information of NMIBC and identified molecular alterations associated with tumour recurrence. A total of 43 patients with NMIBC who underwent transurethral resection of the bladder were enrolled. We performed NGS using an Oncomine panel of tumour specimens and blood samples corresponding to each specimen. The somatic mutation results were analysed by pairwise comparison and logistic regression according to the recurrence of bladder tumours within 1 year. The median incidence of genetic variations in 43 tumour samples was 56 variations per sample, and a high tumour mutation burden (TMB) was associated with tumour recurrence (median variation 33 vs. 64, *p* = 0.023). The most mutated gene was adipose tissue macrophages (*ATM*) (79%), followed by neurofibromatosis-1 (*NF1*) (79%), and neurogenic locus notch homolog protein 1 (*NOTCH1*) (79%). In multivariable analysis, mutation of epidermal growth factor receptor (*EGFR*) (odds ratio [OR], 9.95; 95% confidence interval [CI], 1.40–70.96; *p* = 0.022) and telomerase reverse transcriptase (TERT) (OR, 7.92; 95% CI, 1.22–51.51; *p* = 0.030) were significant factors associated with the recurrence of bladder tumour within 1 year. Our results revealed that high TMB, EGFR mutation, and TERT mutation had a significant association with tumour recurrence in NMIBC. In addition, somatic mutations in EGFR and TERT could be useful prognostic biomarkers in NMIBC.

## 1. Introduction

Bladder cancer is one of the most common cancers worldwide, and its prevalence has increased by 30% in the last decade [1]. The majority of bladder cancers are transitional cell (urothelial) carcinomas, and approximately 75% of patients with bladder cancer present with non-muscle-invasive bladder cancer (NMIBC) [2]. Compared to muscle-invasive bladder cancer (MIBC), NMIBC has significant heterogeneity of different stages and grades and a high cancer-specific survival of 98% for low-grade, non-muscle-invasive cancers [2,3,4]. However, the prognosis of NMIBC is not homogenous despite standard treatment with transurethral resection of bladder tumour (TURBT) followed by adjuvant intravesical therapy. In addition, patients with NMIBC experience a frequent disease recurrence (50–70%) and progression to MIBC (up to 20%) after initial treatment; most patients must undergo long-term cystoscopic surveillance and multiple invasive treatment procedures with TURBT, which make NMIBC one of the most challenging and expensive cancers to diagnose and treat [2,5]. In order to reduce patient and socioeconomic burden and improve post-treatment monitoring in NMIBC, international urological guidelines have stratified disease recurrence and progression according to several clinical and histopathological parameters, but the precise prediction of disease prognosis remains difficult. [2,6] Therefore, more advanced strategies to reveal the risk and predict the prognosis of patients with NMIBC are still needed, leading to growing interest in the genomic evaluation of bladder cancer and identifying accurate non-invasive biomarkers.

Next-generation sequencing (NGS) is a high-throughput method allowing simultaneous sequencing of millions of DNA fragments without previous sequence knowledge. This advanced technology has brought a true revolution by enabling sequencing even with a small amount of DNA fragments using clonal amplification, massive parallel, and cyclic sequencing [7]. In addition, NGS enables the comprehensive assessment of the cancer landscape by sequencing the tumour genome, which helps with disease risk assessment, informative biomarker identification, biomarker-guided genomic-targeted therapy, and the prediction of treatment response and prognosis. Therefore, the use of NGS is rapidly gaining popularity in the clinical practice of other malignancies, such as non-small cell lung, breast, and colorectal cancers [7].

Among urologic malignancies, bladder cancer is a carcinoma with the highest frequency of somatic mutations after skin melanoma and lung cancer [8]. Hence, to date, several studies have evaluated the genomic information of bladder cancer for enabling NGS to be feasible in clinical practice as in other malignancies; however, most have studied MIBC with a less favourable prognosis, and only a few studies have focused on NMIBC due to its significant heterogeneity and high cancer-specific survival [9,10,11,12]. Although large-scale studies have identified recurrent DNA alterations in NMIBC, such as early, abundant mutations in the TERT promoter and chromatin-modifying genes, as well as activating alterations to oncogenes such as FGFR3 [13,14], there is currently a lack of information on the genetic risk factors for disease recurrence and progression of NMIBC. Therefore, in the present study, we evaluated the genomic information of NMIBC and identified somatic molecular alterations associated with tumour recurrence. 

## 2. Materials and Methods

### 2.1. Study Population and Data Collection

All patients who underwent TURBT from January 2017 to December 2020 were selected. Bladder tumours were staged according to the 2017 TNM classification of urinary bladder cancer [15]. Histologic grading of bladder tumours was performed according to the 2004/2016 World Health Organisation (WHO) grading classification system [16]. After confirmation by pathologic examination, patients diagnosed with NMIBC were enrolled in the study. In addition, patients diagnosed with NMIBC were stratified into 4 groups according to the European Association of Urology prognostic-factor risk groups, and patients at intermediate or greater risk of recurrence received adjuvant intravesical Bacillus Calmette–Guerin (BCG) instillation treatment with informed consent. The BCG instillation schedule followed the Southwest Oncology Group method; the 6-week induction course and maintenance course consisted of each week for 3 weeks given at 3, 6, 12, 18, 24, 30, and 36 months from the initiation of induction therapy [17]. Bladder tumour specimens for NGS analysis were obtained at the initial TURBT, and blood samples corresponding to each specimen were collected when patients visited the outpatient department according to their schedule. Patients were excluded if they exhibited any of the following conditions: classification as carcinoma in situ (Tis), bladder tumour specimens not suitable for NGS analysis, histologic diagnosis other than pure transitional cell carcinoma, did not collect blood samples or obtain informed consent, or were lost at follow-up. A total of 43 patients were included in the final analysis. 

Detailed medical histories, including age, body mass index (BMI), sex, diabetes mellitus, hypertension, smoking history, pathologic stage and grade, prognostic-factor risk, adjuvant BCG instillation, and disease recurrence within 1 year, were obtained from the patients’ medical records. Patients were examined by cystoscopy regularly every 3 to 6 months during the first 2 years and biannually up to 5 years. Disease recurrence was defined as a new cystoscopically detected bladder tumour with histologic confirmation after the initial TURBT. 

### 2.2. Patients and Sample Handling

NGS analysis to evaluate the genomic information of NMIBC was performed on the obtained bladder cancer specimens (formalin-fixed paraffin-embedded tissues) from the initial TURBT and blood samples corresponding to each bladder specimen collected from patients. Slides of all bladder cancer specimens were reviewed by one histopathologic specialist. Genomic DNA was extracted from 43 tumour samples and their corresponding blood samples. 

### 2.3. DNA Extraction

Genomic DNA was extracted using the QIAGEN GeneRead DNA FFPE Kit (QIAGEN, Hilden, Germany) following the manufacturer’s instructions. DNA concentration was quantified using the Qubit™ ds DNA High-Sensitive Assay Kit (Thermo Fisher Scientific, Waltham, MA, USA) on the Qubit fluorometer.

### 2.4. Library Preparation and Sequencing

Sequencing was performed using the Oncomine Comprehensive Assay v3 (OCAv3) panel (Thermo Fisher Scientific, Waltham, MA, USA). The OCAv3 panel covers 161 cancer-associated genes, including 87 hotspot genes, 43 focal CNV gains, 48 full CDS for DEL mutations, and 51 fusion drivers, in addition to allowing the detection of single nucleotide variants (SNV), multiple-nucleotide variants (MNV), and small insertions/deletions (indel) [18]. A library was prepared using the OCAv3 kit (Thermo Fisher Scientific, Waltham, MA, USA) according to the manufacturer’s manual. Multiplex PCR was performed using 20 ng of DNA.

The completed library was quantified with a High Sensitivity DNA Kit (Cat. 5067-4626) on a 4200 TapeStation system (Agilent, Santa Clara, CA, USA) and diluted to a final concentration of 14 pM. Diluted samples were subjected to template prep with Ion Chef XL equipment. Template-prepped samples were loaded into the Ion 530™ Chip Kit (Cat. A45850) and analysed on the Ion S5 XL sequencing equipment. Alignment was performed using t-map (v5.10.1). As the reference genome, Hg19 was used.

### 2.5. Data Analysis

The sequenced data were processed using a series of steps. We aligned the sequenced files (FASTQ file) to the reference genome (human reference genome g1k v37) using the BBmap (38.96) and sorting and indexing were performed using Samtools (samtools-1.3.1). Next, filtered alignments are further processed to improve the alignment quality, including local realignment around indels and base quality score recalibration using the Genome Analysis Toolkit (gatk-4.2.6.1). Base quality score recalibration is carried out to recalculate base quality scores for all sequenced reads based on known polymorphisms. The base and mapping quality scores are used to filter reads during variant calling, and the fine-tuning that occurs in this step is important to ensure only high-confidence variants are called.

Variant calling was performed in GATK-Mutect2. Mutect is a method developed for detecting the most likely somatic point mutations in NGS data using a Bayesian classifier approach. We used FilterMutectCalls of the Mutect2 pipeline for a variant filter that filters based on the probability of a somatic variant and optimises the threshold of the “F score” by considering the average of sensitivity and precision. For the detection of somatic single-nucleotide polymorphism (SNP) and insertion and deletion (indel), we used Mutect2 (GATK v4) tumour (bladder tissue) and normal (whole blood) samples (“tumour with matched normal” mode) on each sample. By performing this variant calling, germ-line mutations were excluded, and we analysed oncogene mutations among pure somatic mutations.

The somatic mutation data analyses were carried out using the R4.3.1 package (“maftools”). The mutation results were analysed by pairwise comparison according to the recurrence of bladder tumours within 1 year.

### 2.6. Statistical Analysis

Statistical analysis was performed using the R4.3.1 package (“maftools”). A descriptive analysis was performed to assess patient demographics. Continuous variables are presented as means and standard deviations, and categorical variables are presented as frequencies (%). Clinical and histopathological factors were compared according to the recurrence of bladder cancer within 1 year using Pearson’s χ^2^-test and Fisher‘s exact test for categorical variables and a t-test for continuous variables. A logistic regression test was performed to identify molecular alterations associated with tumour recurrence. *p* < 0.05 indicated statistical significance.

### 2.7. Ethics Statement

The study protocol was reviewed and approved by the institutional review board of the Chonnam National University Hospital (IRB-approved protocol: No. CNUH-2021-205). The study was performed in accordance with the principles of the Declaration of Helsinki and the Ethical Guidelines for Clinical Studies. Informed consent was obtained from all patients. 

## 3. Results

The characteristics of the patients who received transurethral resection for NMIBC are summarised in Table 1. The mean age and BMI of the enrolled patients were 70.6 ± 9.6 years and 25.2 ± 3.3 kg/m^2^, respectively. Thirty-seven (86.0%) patients were male, 25 (58.1%) patients had hypertension, 12 (27.9%) patients had diabetes mellitus, and 25 (58.1%) patients had a smoking history. Regarding histopathologic results, the numbers of patients with Ta-low grade, Ta-high grade, or T1-high grade were 31 (72.1%), 11 (25.6%), and 1 (2.3%), respectively. According to the European Association of Urology prognostic-factor risk groups, 14 (32.6%) patients were low-risk, 21 (48.8%) patients were intermediate-risk, and 8 (18.6%) patients were high-risk. Twenty-four (55.8%) patients were administered adjuvant intravesical BCG treatment as part of a planned schedule. During the follow-up period, 22 patients (51.2%) had disease recurrence within 1 year regardless of adjuvant treatment and additionally received treatment with TURBT. Of 22 patients with disease recurrence within 1 year, 17 (77.3%) patients had Ta-low-grade recurrence, and there was no disease progression to T2. Regarding comparisons of clinical and histopathological features according to the recurrence of bladder cancer within 1 year, there were no significant differences between the two groups except smoking history (Table 2).

The results of analysing somatic mutations in tumour tissues and blood samples corresponding to each bladder specimen are presented in Figure 1. The median incidence of genetic variations in 43 tumour samples was 56 variations per sample, and a frameshift deletion was the most common variant classification. Among the six single nucleotide variation (SNV) classes, C > T was the most common SNV pattern. 

The most mutated gene was ATM (79%), followed by NF1 (79%) and NOTCH1 (79%). High TMB was associated with tumour recurrence (median variation 33 vs. 64, *p* = 0.023) (Appendix A). However, none of the top 10 mutated genes had a significant difference in a pairwise comparison according to the recurrence of bladder tumours within 1 year, as shown in Figure 2. However, EGFR mutation (3 vs. 10, *p* = 0.026) and TERT mutation (4 vs. 11, *p* = 0.033) were significantly different in further pairwise comparison analyses (Figure 3).

The predictive factors associated with tumour recurrence are shown in Table 3. Univariate analyses revealed a significant association of TMB, EGFR mutation, and TERT mutation with tumour recurrence within 1 year (*p* = 0.030, *p* = 0.033, and *p* = 0.039, respectively). In addition, multivariable analyses showed somatic mutations in EGFR (odds ratio [OR], 9.95; 95% confidence interval [CI], 1.40–70.96; *p* = 0.022) or TERT (OR, 7.92; 95% CI, 1.22–51.51; *p* = 0.030) were significant factors associated with the early recurrence of NMIBC. 

## 4. Discussion

Although NMIBC has a significantly higher cancer-specific survival rate compared to MIBC, patients with NMIBC have a highly diverse clinical course according to histopathologic and patient-related factors [2,3,4]. In addition, some patients with NMIBC experience frequent tumour recurrence despite receiving standard treatment with TURBT followed by adjuvant intravesical therapy, which leads to a significant patient and socioeconomic burden [5]. Therefore, there is a growing interest in more advanced strategies to reveal risk and predict prognosis, and several physicians have focused on the genomic evaluation of bladder cancer. Thus, we evaluated the genomic information of NMIBC and identified molecular alterations associated with tumour recurrence. We found that EGFR mutations or TERT mutations might have prognostic value for the tumour recurrence of NMIBC.

Recently, the rapid development of NGS technology has enabled us to identify comprehensive genetic information related to cancer, and leveraging genomic data based on NGS analysis has improved disease risk assessment, genomic-targeted therapy, and the prediction of treatment responses and prognosis [7]. Bladder cancer, especially NIMBC, has significant heterogeneity with different stages, grades, and genomic alterations [2,3,4,8]. Therefore, many physicians have made an effort to evaluate the relationship between genomic alterations and NMIBC. Indeed, recently, the tumour mutation burden (TMB) was reported to be an emerging biomarker for the prediction of tumour behaviour. Jia Lv et al. analysed the clinical significance of TMB in 443 bladder cancer samples obtained from The Cancer Genome Atlas (TCGA) and found that TMB was related to high-grade bladder cancer [19]. In the present study, the median incidence of genetic variations in 43 tumour samples was 56 variations per sample, and a high TMB was associated with tumour recurrence, which is consistent with previous reports. In addition, previous studies reported that patients with a high TMB had better overall survival and response to immunotherapy compared with those with a low TMB [19,20]. However, in the present study, overall survival was not analysed because NMIBC has a high cancer-specific survival rate.

The most frequently mutated genes in bladder cancer samples in the present study were ATM, NF1, and NOTCH1. ATM is a protein-coding gene that encodes a PI3K-related serine/threonine protein kinase that helps maintain genomic integrity. ATM gene mutations are commonly associated with increased predisposition to various cancers (more often in hematologic malignancies) and poor prognosis because ATM has a crucial role in the repair of DNA double-strand breaks [21]. The NF1 gene, highly related to neurofibromatosis, encodes neurofibromin, a tumour suppressor protein that acts as a negative regulator of the Ras/MAPK and PI3K/mTOR signalling pathways [22]. The Notch signalling pathway relies on cell–cell contact to influence cellular behaviour, and it has been implicated in human cancer as an oncogene and tumour suppressor [23]. Based on previous reports, mutations of ATM, NF1, and NOTCH1 in bladder cancer are found in approximately 15%, 10%, and 50% of cases, respectively [21,23,24]. These results are inconsistent with those of the present study. In addition, previous large-scale studies of NMIBC have identified early, general mutations in the TERT promotor and chromatin-modifying genes such as KDM6A and ARID1A, as well as activating alterations to oncogenes such as FGFR3 and ERBB2 [13,14,20]. In the present study, these gene mutations were not included in the top 10 mutated genes, except for ARID1A. (ARID1A 67%, FGFR3 47%, TERT 35%, and ERBB2 23%) This may be due to the number of targeted genes included in the gene panel and the small size of our study population. Therefore, further research on the relationship between these gene mutations and NMIBC is needed in the future. 

In the present study, the mutation results were analysed by pairwise comparison according to the recurrence of bladder tumours within 1 year. The results revealed that the EGFR mutation and the TERT mutation had a significant association with tumour recurrence. In addition, multivariable analyses showed that EGFR and TERT mutations were significant factors associated with the recurrence of NMIBC. EGFR is the product of the c-erbB1 proto-oncogene, which belongs to the tyrosine kinase receptor family. It is a receptor for several growth factors, such as epidermal growth factor, transforming growth factor alpha, amphiregulin, heparin-binding EGF-like factor, betacellulin, and epiregulin [25]. EGFR activation causes transcriptional activation and cell proliferation via intracellular cascade events. Indeed, several studies reported an association between EGFR overexpression and malignancies, such as lung, breast, and colon cancers [26,27]. Regarding bladder cancer, the results of previous studies have revealed that more than half of cases of bladder cancer had EGFR overexpression, which was associated with high tumour grade, stage, tumour progression, and poor prognosis [25,28]. These results are consistent with those of the present study, which strongly suggest that EGFR might be a useful prognostic biomarker in bladder cancer, even in NMIBC. 

Activating mutations in the TERT gene increase telomerase expression, allowing some neoplasms to overcome the end-replication problem and avoid senescence [29]. TERT mutations have the highest prevalence in glioblastoma, liposarcoma, oligodendroglioma, urothelial carcinoma, melanoma, and hepatocellular carcinoma [30]. Regarding bladder cancer, TERT mutations are an early event in bladder tumourigenesis, and a recent study reported the potential of TERT mutations as a promising non-invasive biomarker for the early detection of bladder cancer [31]. In the present study, our results showed that TERT might be a potential biomarker for tumour recurrence. In addition, our results may support the importance of TERT mutations in NMIBC recurrence, and further studies will be needed to validate these findings and evaluate their clinical application. 

The present study had some limitations. First, this was a small-scale study and only included patients from a single institution. Further studies using more bladder cancer samples from multiple institutions are needed to identify the precise role of gene mutations in bladder cancer recurrence. Second, our targeted sequencing approach for cancer sample tissues can be limited by biopsy bias, and it is difficult to evaluate precise evolutionary relationships over time. Third, we did not consider the effect of adjuvant intravesical BCG treatment. As BCG can provide temporal changes in the genomic environment of early bladder cancer [20], further studies on recurrent bladder cancer samples are needed to account for this aspect. However, our results indicate that ancestor gene mutations are associated with NMIBC recurrence. In addition, considering the results of this study, current strategies for the surveillance of bladder cancer via macroscopic cystoscopy and urine cytology can miss the possibility of residual disease, so including a genetic evaluation should be considered for the stratification of recurrent risk.

## 5. Conclusions

The clinical application of NGS-based analysis is already accepted as an essential step towards precision oncology. NGS-based analysis and leveraging genomic data are anticipated to improve the classification of bladder cancer patients with recurrent risk. Considering this trend, our results may support the importance of somatic mutations in NMIBC recurrence and the fact that somatic mutations in EGFR and TERT have a prognostic value for tumour recurrence in NMIBC. However, additional research to validate these findings and evaluate their clinical application is needed. 

## Figures and Tables

**Figure 1 diagnostics-13-03201-f001:**
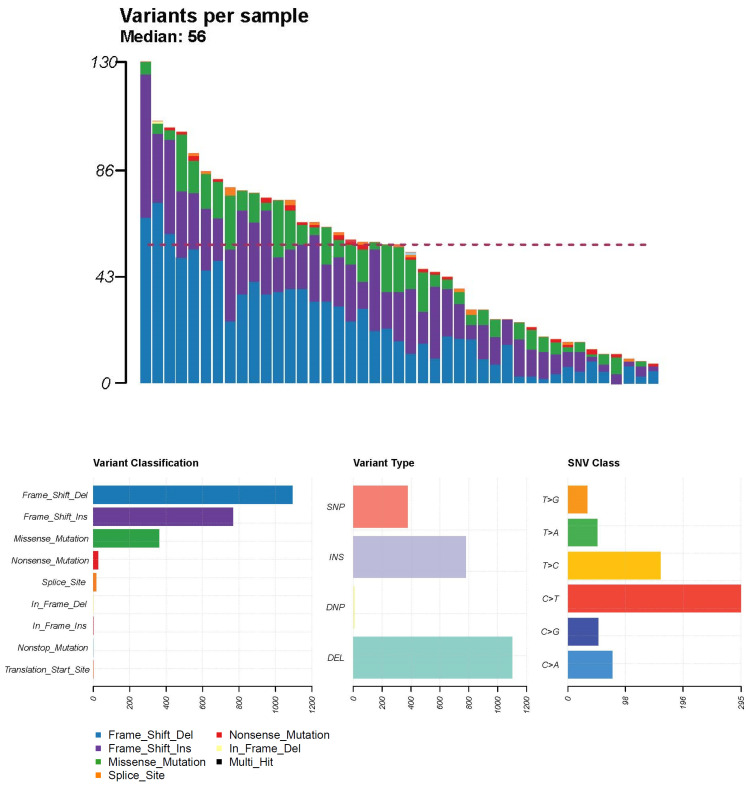
Results of somatic mutations in tumour tissues and blood samples corresponding to each bladder specimen.

**Figure 2 diagnostics-13-03201-f002:**
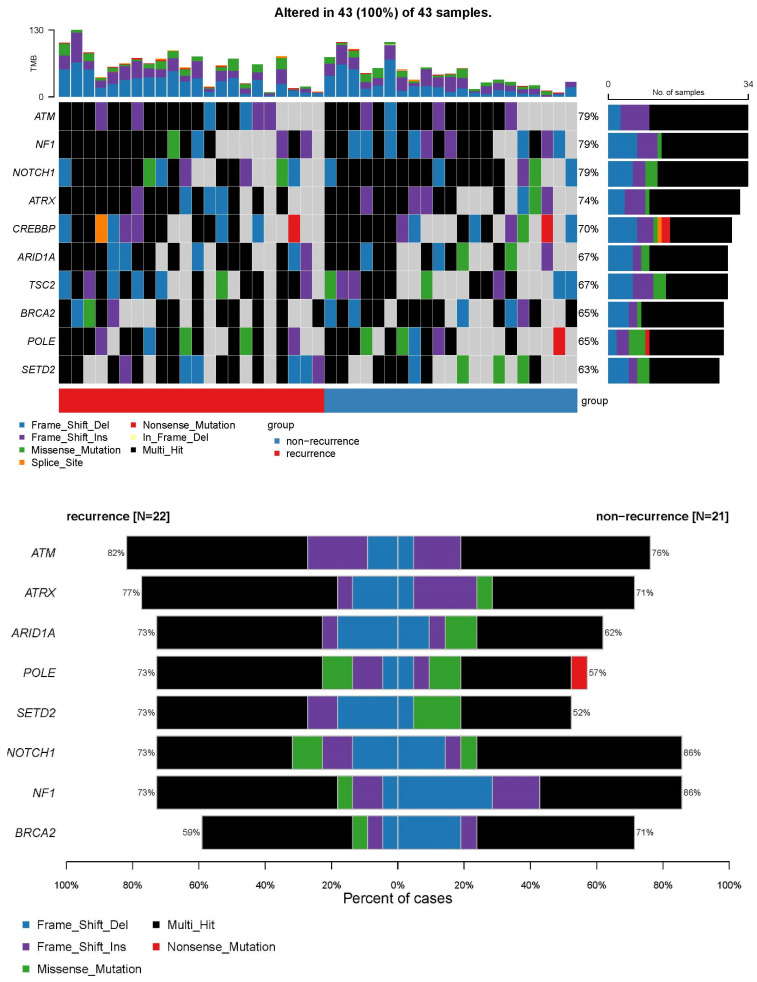
Results of pairwise comparison according to the recurrence of bladder cancer within 1 year.

**Figure 3 diagnostics-13-03201-f003:**
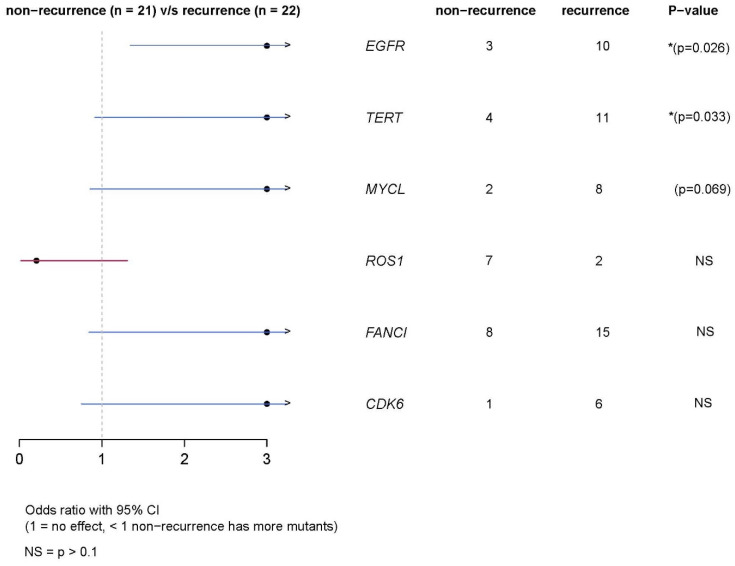
Results of somatic molecular alterations associated with the early recurrence of bladder cancer.

**Table 1 diagnostics-13-03201-t001:** Characteristics of patients who received transurethral resection for non-invasive bladder cancer.

Variables	Values (*N* = 43)
Age (years)	70.6 ± 9.62
BMI (kg/m^2^)	25.2 ± 3.27
Sex	
Male	37 (86.0)
Female	6 (14.0)
Comorbidities	
HTN	25 (58.1)
DM	12 (27.9)
Smoking history	25 (58.1)
Pathologic stage and grade	
Ta, low grade	31 (72.1)
Ta, high grade	11 (25.6)
T1, high grade	1 (2.3)
EAU prognostic-factor risk	
Low	14 (32.6)
Intermediate	21 (48.8)
High	8 (18.6)
Adjuvant BCG instillation	
Yes	24 (55.8)
No	19 (44.2)
Recurrence within 1 year	
Yes	22 (51.2)
No	21 (48.8)

BMI: body mass index; DM: diabetes mellitus; HTN: hypertension; EAU: European Association of Urology; BCG: Bacillus Calmette–Guerin. Data are represented as the mean ± standard deviation, or *n* (%).

**Table 2 diagnostics-13-03201-t002:** Comparisons of clinical and histopathological features according to the recurrence of bladder cancer within 1 year.

Variable	Recurrence within 1 Year		*p*-Value
	Yes (*n* = 22)	No (*n* = 21)	
Age (years)	72.09 ± 8.60	69.05 ± 10.58	0.306
BMI (kg/m^2^)	25.15 ± 3.27	25.18 ± 3.23	0.280
Sex			1.000
Male	19 (86.4)	18 (85.7)	
Female	3 (13.6)	3 (14.3)	
Hypertension	12 (54.6)	13 (61.9)	0.625
Diabetes mellitus	5 (22.7)	7 (33.3)	0.438
Smoking history	16 (72.7)	9 (42.9)	0.047
Pathologic stage and grade			0.052
Ta, low grade	13 (59.1%)	18 (85.7%)	
Ta and T1, high grade	9 (40.9%)	3 (14.3%)	
EAU prognostic-factor risk			0.159
Low	5 (22.7%)	9 (42.9%)	
Intermediate + high	17 (77.3%)	12 (57.1%)	
Adjuvant BCG instillation			0.658
Yes	9 (40.9%)	10 (47.6%)	
No	13 (59.1%)	11 (52.4%)	

BMI: body mass index; DM: diabetes mellitus; HTN: hypertension; EAU: European Association of Urology; BCG: Bacillus Calmette–Guerin. Data are represented as the mean ± standard deviation, or *n* (%).

**Table 3 diagnostics-13-03201-t003:** Factors associated with the early recurrence of bladder cancer.

Variable	Univariate Analysis		Multivariate Analysis	
	Odds Ratio (95% CI)	*p*-Value	Odds Ratio (95% CI)	*p*-Value
Age	1.03 (0.97–1.10)	0.300		
BMI (≥25)	1.95 (0.58–6.58)	0.282		
Sex (female)	0.95 (0.17–5.32)	0.951		
Hypertension	0.74 (0.22–2.49)	0.625		
Diabetes mellitus	0.59 (0.15–2.26)	0.440		
Smoking history	3.56 (0.99–12.73)	0.051		
Histologic grade				
Low	Reference			
High	4.15 (0.94–18.41)	0.061		
EAU prognostic risk				
Low	Reference			
Intermediate, high	2.55 (0.68–9.54)	0.164		
Adjuvant BCG treatment	0.76 (0.23–2.54)	0.658		
EGFR mutation	5.00 (1.14–22.02)	0.033	9.95 (1.40–70.96)	0.022
TERT mutation	4.25 (1.08–16.77)	0.039	7.92 (1.22–51.51)	0.030

CI: confidence interval; BMI: body mass index; EAU: European Association of Urology; BCG: Bacillus Calmette–Guérin; EGFR: epidermal growth factor receptor; TERT: telomerase reverse transcriptase.

## Data Availability

The data presented in this study are available in this article.

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
