# Peer review of "Somatic Mutation of the Non-Muscle-Invasive Bladder Cancer Associated with Early Recurrence"

_diagnostics, 2023, doi:10.3390/diagnostics13203201_

Round 1

Reviewer 1 Report

This is a good study of a small cohort of patients with diverse T-stage/risk groups of NMIBC.  The findings are not consistent with previously published literature, but the authors discuss this very well. This cohort needs to be published to provide a counter point to existing literature, however, some improvements can be made.

Do the authors have smoking status available, and can this be included in the manuscript?

Can the authors analyse molecular profie/NGS findings for the different pathologic stage/grade, as well as the different EAU risk groups?  I would have expected a higher rate of FGFR in the low/intermediate groups.  It would provide value to split the findings in these categories.  

Do you have data on progression as well as recurrence?  Were most recurrences, low grade recurrences?  More detail would be helpful.

Author Response

I attach the file for response to the reviewer`s comments.

Reviewer 2 Report

Non-muscle invasive bladder cancer(NMIBC) is well known for its frequent recurrence, which requires lengthy and costly long term surveillance and treatment. Study somatic mutation of NMIBC could discover potential genetic driver mutations leading to the recurrence and design genetic detection method for the recurrence. Your study sequenced genetic mutations using Oncoming Comprehensive Assay panel in the Ion S5 XL sequencing equipment (Thermo FISHER Scientific) of 43 NMIBC in TURBT specimens. You have found that high tumor mutation burden(TMB), EGFR and TERT mutation were associated with recurrence within one year, which is compatible with previous study results of invasive bladder cancer. 

However, in Figure 2 of your result, there are extremely high mutation rates of genes of ATM, NF1, POLE and BRC2 detected in this group of NMIBC, which is not possible scientifically as they are 10 to 20 times higher compared with previous reports. The problem maybe due to over-sensitivity of the Thermo Fisher sequencing platform or contamination of the specimens. Please find out the cause of the extremely high rate of the gene mutation,  adjust or refine the result/data.  

Author Response

(The authors gave the same response as above.)

Reviewer 3 Report

Dear Author

Manuscript ID: diagnostics-2587484

The manuscript has some issues that should be solved before publication

1-      Both blood sample and TURBT tissue were taken for NGS and it is not clear which one is used. There is not clear that genomic DNA from whole blood sample is sequenced which include germ line mutations or bladder tissue samples is sequenced which represent somatic mutations.

2-      It is interesting that author found 56 variations per sample, and a frameshift 185 deletion was as the most common variant. Among the six single nucleotide 186 variation (SNV) classes, C > T was the most common SNV pattern. I am interested in knowing the name of variant and mutations.

3-      The result indicated that (ATM) (79%), (NF1) (79%), and 23 neurogenic locus NOTCH1 (79%),  24 EGFR and TERT were associated with non-muscle invasive bladder cancer (NMIBC) tumor recurrence after one year. However, conclusion indicated to that TMB, EGFR mutation, and TERT mutation had a significant association with 28 tumor recurrence of NMIBC. How the author finally delete NF1, NOTCH1? The TMB is not the name of gene. TMB is the tumor mutation burden is the number of changes (called genetic mutations) found in the DNA of cancer cells.

4-      The treatment strategy for these 43 patients with 16 NMIBC is not clear (total cystectomy, chemo-radiotherapy, partial cystectomy)

Author Response

(The authors gave the same response as above.)

Round 2

Reviewer 2 Report

I am satisfied with your explanation and answer to my comment.